# A Two-for-One Diagnosis: A Rare Case of Chronic Abdominal Pain Caused by Gastroptosis and Wilkie’s Syndrome in a Young Woman

**DOI:** 10.3390/diagnostics15030270

**Published:** 2025-01-23

**Authors:** Nosseir Youssoufi, Ayoub Jaafari, Sohaïb Mansour, Mohamed El Hamdi, Andrea Gallerani, Charalampos Pierrakos, Rachid Attou

**Affiliations:** 1General Medicine and Emergency Care Department, CHU Brugmann, 1020 Brussels, Belgium; nyoussou@hotmail.com; 2Nuclear Medicine and Intensive Care Department, CHU Brugmann, 1020 Brussels, Belgium; 3Cardiology and Intensive Care Department, CHU Saint-Pierre, 1000 Brussels, Belgium; sohaib.mansour@ulb.be; 4Gastroenterology Department, CHIREC Hospital Group, 1160 Brussels, Belgium; mohamed.elhamdi@chirec.be; 5Intensive Care Deparment, CHU Brugmann, 1020 Brussels, Belgium; andrea.gallerani@chu-brugmann.be (A.G.); charalampos.pierrakos@chu-brugmann.be (C.P.); rachid.attou@chu-brugmann.be (R.A.)

**Keywords:** gastroptosis, visceroptosis, Glenard’s disease, Wilkie’s syndrome, superior mesenteric artery syndrome, SMAS, chronic abdominal pain

## Abstract

Long-term abdominal pain (LAP) affects 30% to 40% of children, often linked to functional gastrointestinal disorders (FGIDs) such as functional dyspepsia and irritable bowel syndrome. Less common causes include gastroptosis and superior mesenteric artery (SMA) syndrome, conditions that can be challenging to diagnose due to their rarity. Gastroptosis refers to the downward displacement of the stomach, while SMA syndrome, also known as Wilkie’s syndrome, involves the compression of the duodenum between the abdominal aorta and the superior mesenteric artery. While both conditions have been described separately, their coexistence has not been previously documented. Herein, we present the case of a 17-year-old girl with a six-month history of postprandial abdominal pain and vomiting, diagnosed with both gastroptosis and SMA syndrome. Diagnostic tests, including a CT scan and barium radiography, confirmed the presence of a duodenal stricture and severe gastric elongation, providing an insight into the pathophysiology of these rare conditions.


Figure 1Abdominal CT scan with injection of contrast medium showed a duodenal stricture on a mesenteric pinch (red arrow) between the aorta (red star) and the superior mesenteric artery (blue star) on sagittal (**A**–**C**) and axial (**D**) sections. We present the case of a 17-year-old female patient who presented to her general practitioner and the gastroenterology department with her parents due to epigastric pain, nausea, vomiting, and significant progressive weight loss (11 kg in 6 months). The patient reported a sensation of heaviness and abdominal discomfort, especially after meals, accompanied by nausea and vomiting. These symptoms had been present for 6 months, but had become progressively more intense over the last 3 months. It is noteworthy that the epigastric pain (rated up to 6/10 by the patient), nausea, and vomiting (emerging 30 min to 1 h after meals) had escalated in intensity over the past three months, reaching a point where the patient found them unbearable and feared that further weight loss would exacerbate her condition. She reported that her appetite remained relatively stable, but her food intake had gradually decreased from four meals a day to three and then to two. There had been no changes in her diet, as confirmed by her parents, and she maintained a healthy lifestyle. She had made several visits to emergency departments for her symptoms and had tried various pharmacological agents, including proton pump inhibitors (PPIs) and antispasmodics, in an attempt to alleviate her condition. However, these interventions yielded only minimal and transient symptomatic improvement. The rest of her medical history was unremarkable, with no history of fever, dysphagia (solid or liquid), transit disorders, constipation or diarrhoea, chest pain, or dyspnoea. Her medical and surgical history was devoid of any noteworthy abnormalities. The family history was unremarkable. The infant was delivered via spontaneous vaginal delivery at 38 weeks of gestation with a birth weight of 2750 g. The infant was breastfed until the age of 1.5 months, exhibited normal growth and development, and was up to date with her vaccinations. The patient’s clinical parameters at the time of consultation were found to be entirely satisfactory, with the following observations: blood pressure of 120/64 mmHg, heart rate of 74 bpm, body temperature of 36.1 °C, and an oxygen saturation of 100% on room air. The subject’s weight was 45 kg with a height of 164 cm, resulting in a body mass index of 16.7 kg/m^2^. On clinical examination, cardiopulmonary auscultation was within normal limits, and an abdominal examination revealed epigastric tenderness radiating to the umbilicus, but no associated mass or rebound. Peristalsis of the digestive tract was noted to be intact. Given the circumstances, additional diagnostic procedures were initiated. Laboratory investigations revealed no haemogram abnormalities, no acute and/or chronic inflammation (C-reactive protein (CRP) and erythrocyte sedimentation rate (ESR) negative), preserved renal function, a satisfactory ionogram, normal thyroid function (TSH and fT4 N), and normal phospho-calcium balance (PTH and 25-OHD normal). A test for celiac disease was negative (total IgA and anti-transglutaminase antibodies were normal). Faecal calprotectin (FC) levels were also normal (<100 µg/g, N < 100 µg/g), ruling out inflammatory bowel disease (IBD). An endoscopic examination of the stomach revealed erythematous gastritis, with no additional microscopic abnormalities identified. An abdominal computer tomography scan (CT scan) with injection of contrast medium demonstrated a duodenal stricture over a mesenteric pinch (Figure 1, red arrow). Furthermore, a barium radiographic examination was carried out and showed a severe elongation of the stomach down to the umbilicus, suggestive of gastroptosis (Figure 2, white arrows). The patient was diagnosed with gastroptosis associated with superior mesenteric artery syndrome.
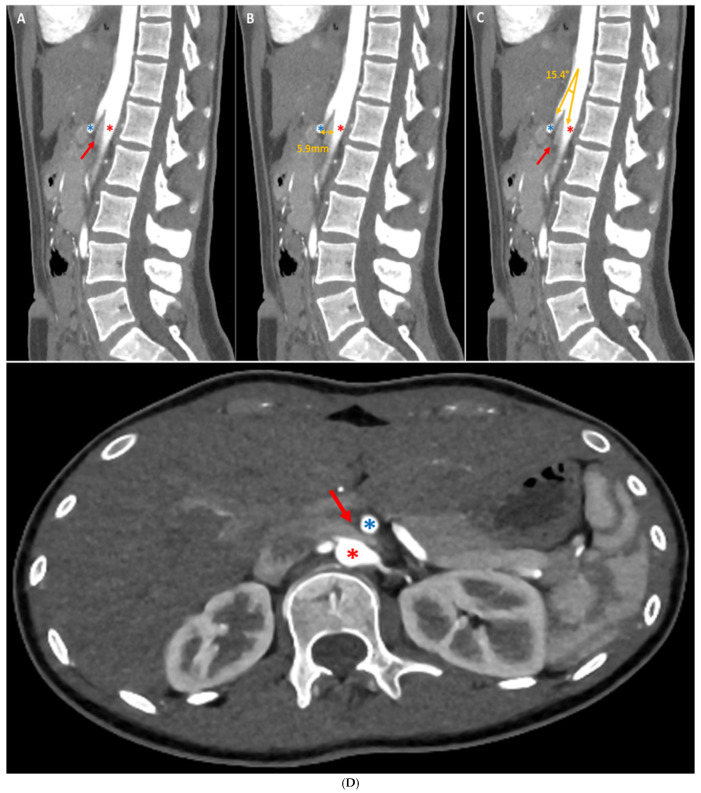

Figure 2Fluoroscopic examination of the oesophagus and digestive tract with barium revealing severe elongation of the stomach at the umbilicus throughout the dynamic passage of the contrast medium (**A**–**D**), suggestive of gastroptosis (white arrows). The term “chronic abdominal pain” is defined as pain that persists or recurs over a period of more than three months, or that lasts for more than three months [1]. It is a pervasive issue in the paediatric population, representing a significant contributor to both inpatient and outpatient consultations across a spectrum of healthcare settings, including general practices, paediatric wards, gastro-enterology, and emergency departments [1]. This condition has a significant impact on children’s quality of life and may cause considerable stress and anxiety for their parents. As some recent epidemiological studies indicate, the prevalence is estimated between 22% and 25% in the United States and Western countries, with a higher prevalence among women than men [1,2]. In 1833, French physician Frantz Glenard initially proposed the term “visceroptosis” in his doctoral thesis, defining it as a prolapse or sagging of the abdominal viscera (internal organs) below their normal position resulting from ligament laxity, and suggesting it as a potential cause of chronic abdominal pain [3]. When the stomach is affected more specifically, it is defined as gastroptosis. Concurrently, Rokitansky [4] proposed a different entity, also responsible for chronic abdominal pain, which he described as a phenomenon of obstruction and dilatation of the proximal intestine caused by compression of the third portion of the duodenum by the superior mesenteric artery (SMA) or, more specifically, by the SMA and the aorta [4]. The condition, initially referred to as either “internal hernia” or “internal incarceration”, received its first comprehensive account from David Wilkie in 1921. This detailed description led to the establishment of Wilkie’s syndrome, also known as superior mesenteric artery syndrome (SMAS) [4]. Although isolated cases have been documented in the literature, no one has hitherto reported these rare conditions simultaneously in the same patient, as in our case [5,6]. In SMAS, the data currently available show that women are mainly affected in the 20–50 age group, with a higher prevalence in women than in men (ratio 3:2) [7]. A number of risk factors appear to be involved in this phenomenon. Among those identified are severe underweight due to certain catabolic conditions (such as cancer, neurologic injury, surgery, burns, malabsorption syndromes, trauma, or psychiatric problems), postural defects (such as congenital scoliosis), and excessive laxity of the abdominal wall and mesenteric attachments, which become too thin and relaxed under the weight of the organ [5,7,8]. The clinical manifestations are frequently numerous but not particularly specific, including a lengthy history of intermittent postprandial abdominal discomfort, bloating, or epigastric distress, recurrent episodes of nausea and vomiting, early satiety, and weight loss, all of which are exacerbated by standing or postprandial activity [1,5,6,8]. The aetiology of abdominal pain is a complex phenomenon, with no single causal model proving wholly adequate. Indeed, gastroptosis and SMAS may overlap considerably more common gastrointestinal disorders including gastritis; peptic ulcer disease; pancreatitis; pancreatic tumors (above a certain size); gastro-oesophageal reflux disease (GERD); inflammatory bowel diseases, such as Crohn’s disease affecting the upper gastrointestinal tract; celiac disease; and irritable bowel syndrome [5,6,7,8]. Additionally, less frequent yet equally pertinent conditions that may elucidate the symptomatology were discussed within the context of anatomical conditions, including congenital stenosis of the duodenum and intestines, intestinal malrotation, food allergies, and psychosomatic disorders [5]. It is evident that all the aforementioned causes were excluded in our patient. Consequently, the absence of Crohn’s disease/RCUH or any intestinal infection was confirmed (CRP, ESR, and faecal calprotectin negative), as was the absence of pancreatic disorders (normal liver and pancreatic enzymes), and the absence of celiac disease (total IgA and anti-transglutaminase antibodies normal) or a food allergy (absence of skin reactions and IgE negative). A gastroscopy was performed, revealing only discrete erythematous gastritis, which did not fully explain the underlying clinical condition of the patient. Consequently, abdominal imaging was conducted to exclude less common etiologies, including congenital stenosis of the duodenum and intestines, and intestinal malrotation. This comprehensive diagnostic approach enabled the formulation of a definitive diagnosis. As evidenced by the existing literature, imaging has a crucial role in the diagnostic process of gastroptosis and SMAS [9]. Fluoroscopy with oral barium or iodine contrast remains the primary assessment tool of the upper gastrointestinal tract, offering a real-time dynamic visualization that facilitates the correlation of clinical history with imaging [5,8,9]. In both case, fluoroscopic findings generally include gastrointestinal distention (Figure 2, white arrows where the stomach is displaced downwards, with the greatest curvature close to the iliac crest, while the antrum remains in place), delayed gastric emptying (Figure 2, dynamic A–D), and a narrowing of the third part of the duodenum (less obvious in our case). Further information can be obtained using computed tomography (CT), which, in combination with intravascular contrast, allows a detailed assessment of the aorto-mesenteric vascularization and caliber of the gastroduodenal structures, proving invaluable in the diagnosis. SMAS syndrome is characterized by gastroduodenal dilatation with a marked narrowing at the junction of the AMA and the aorta (Figure 1, red arrow), shortening of the aorto-mesenteric distance (normal: 10–34 mm; Figure 1B), and tightening of the aorto-mesenteric angle (normal: 28–65°; Figure 1C) [8,9]. Gastroptosis may also be visible, depending on whether or not the patient has eaten prior to the examination; however, it may be less visible if the examination is performed on an empty stomach. According to our patient, the various symptoms can be explained by our two concomitant pathologies. Firstly, the epigastric pain and sensation of heaviness, especially after meals, may be linked to the gastroptosis, and can lead to delayed gastric emptying and impaired motility, which probably exacerbates postprandial discomfort. The elongation of the stomach to the umbilicus, as demonstrated by the barium study (Figure 2), signifies a condition that interferes with the normal physiological position of the stomach, resulting in a sensation of fullness and heaviness after eating. Secondly, the nausea and vomiting, which are progressive over time, can be explained by SMAS (Wilkie’s syndrome). In SMAS, the third portion of the duodenum is compressed by the superior mesenteric artery due to a reduced angle between the artery and the aorta, often caused by the loss of mesenteric fat or an abnormal position of the stomach. The duodenal stenosis seen on the CT scan (Figure 1, red arrow) corresponds to this pathophysiology. The consequence of this compression is the impairment of the function of the duodenum, which can result in symptoms such as nausea, vomiting, and postprandial discomfort. Furthermore, the stricture of the duodenum contributes to delayed gastric emptying, which in turn prevents the progression of food through the gastrointestinal tract, thus causing vomiting. Finally, the significant and progressive weight loss experienced by the patient can be attributed to a combination of chronic nausea, vomiting, and poor nutrient absorption. Gastroptosis has been demonstrated to impair stomach function, leading to malnutrition due to inadequate digestion, a condition that is exacerbated by vomiting, thereby preventing adequate calorie intake. In addition, the duodenal compression in SAMS probably impairs nutrient absorption, contributing to further weight loss. In the current clinical practice, invasive treatment is reserved for a limited number of cases complicated by intestinal obstruction [10]. The treatment of gastroptosis and SMAs is primarily focused on the management of the symptoms and correction of the underlying anatomical abnormalities. For gastroptosis, conservative management encompasses dietary modifications, postural adjustments, and physical therapy to strengthen the abdominal and spinal musculature. Surgical interventions, such as gastroplication or gastropexy, are employed in severe cases [5,7,8,10]. For SMAS, it is primarily based on nutritional support, including the administration of high-calorie diets or enteral feeding to facilitate weight gain and relieve vascular compression. Additionally, postural modifications and, in certain cases, prokinetic drugs may be employed. In instances where conservative measures prove ineffective, surgical options, such as duodenojejunostomy or SMA release surgery, may be necessary to address the underlying anatomical issues [7,8,9]. The management of both conditions requires a multidisciplinary approach, involving the expertise of gastroenterologists, nutritionists, and surgeons, tailored to the specific needs of each patient. In conclusion, gastroptosis and Wilkie’s syndrome are two rare entities that are underdiagnosed and rarely reported in the literature, representing a diagnostic and therapeutic challenge in clinical practice. Our case highlights the necessity for a multidisciplinary approach involving a general practitioner, gastroenterologists, nutritionist, radiologists, and surgeons to accurately diagnose and treat these overlapping conditions. A comprehensive history, thorough physical examination, and appropriate imaging studies are crucial to circumvent such diagnostic pitfalls.
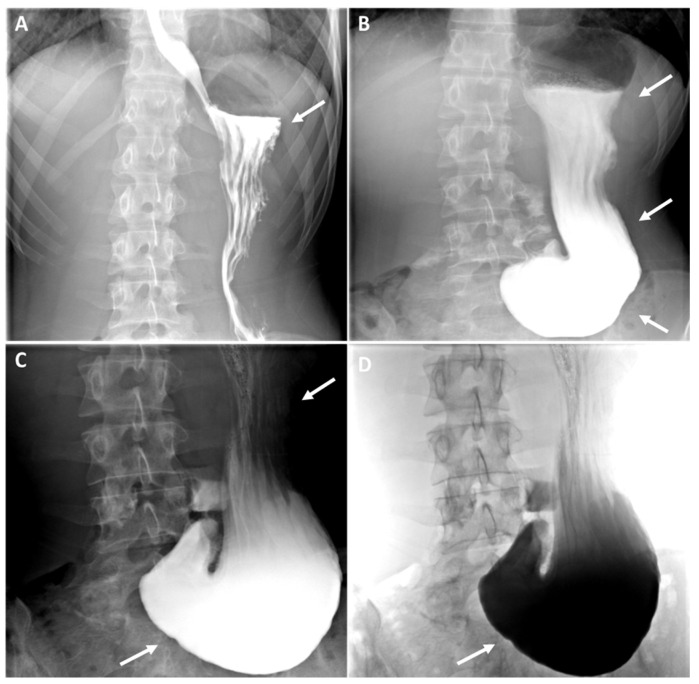



## Data Availability

The data used and analyzed in this study are available from the corresponding author on reasonable request.

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
