# Peer review of "A Two-for-One Diagnosis: A Rare Case of Chronic Abdominal Pain Caused by Gastroptosis and Wilkie’s Syndrome in a Young Woman"

_diagnostics, 2025, doi:10.3390/diagnostics15030270_

Round 1
Reviewer 1 Report
Comments and Suggestions for Authors
Dear Authors,
I have read an interesting case of gastroptosis and SMA syndrome. The manuscript could benefit from two things:
1) Please consider limiting the references to ten or less
2) As for the description of figure 1 and 2, please describe the figures accordingly. Also, is there a coronal CT for the gastroptosis?
Author Response
Greetings, and gratitude is extended for your consent to evaluate our article. Your feedback has been given due consideration, resulting in the creation of a revised version that incorporates a reorganization of the sources (up to ten) and a clarification of the figures presented.
It should be noted that coronal images are indeed present in Figure 1; however, following a re-reading of the article in conjunction with our radiology colleagues, it was determined that these images are not particularly beneficial. This is due to the presence of evidence of duodenal narrowing by the aorta and superior mesentery, which is more clearly visualised in the sagittal and axial sections.
It is hoped that this version will be received positively, and that the article will be published in the near future in MDPI's Diagnostic.
Warm Regards
Dr. Jaafari Ayoub

Reviewer 2 Report
Comments and Suggestions for Authors
After careful reading of this manuscript I found that it presents a unique case of a 17-year-old girl diagnosed with both gastroptosis and Wilkie's syndrome, given the challenges associated with diagnosing chronic abdominal pain in paediatric patients.
However there are some revisions which should be integrated:
- Keywords should be more specific includes Wilkie's syndrome and Glenard's disease.
- One important point clinically is the lack of timeline of symptom progression to better illustrate the clinical course. It should be integrated
- The authors have mentioned that common gastrointestinal disorders were excluded through laboratory investigations and imaging. Authors should explain how these exclusions were made, particularly regarding any specific tests conducted.
- While the authors discuss the findings from imaging studies, a more explicit correlation between these findings and the clinical symptoms experienced by the patient would provide deeper insights into the pathophysiology of gastroptosis and Wilkie's syndrome.
Author Response
Greetings, and gratitude is extended for the time invested in providing feedback on our article.
We have given due consideration to your comments and have endeavoured to make the temporality of our patient's symptoms clearer, as well as providing a clearer explanation of the correlation between the symptoms, clinical examinations and diagnosis.
It is our sincere hope that these revisions have met your expectations.
It is our hope that our article will be published in this journal.
Warm Regards
Dr Jaafari Ayoub

Reviewer 3 Report
Comments and Suggestions for Authors
A review of the article:
A 2-for-1 diagnosis: A rare case of chronic abdominal pain 2 caused by gastroptosis and Wilkie’s syndrome in a young woman.
The authors presented a case of particular interest: a 17-year-old female patient diagnosed with both gastroptosis and superior mesenteric artery (SMA) syndrome. These two conditions are seldom reported together, and the case emphasizes the importance of a multidisciplinary approach to diagnosis and treatment.
In the abstract, brevity is important, the introduction should be brief and the authors should focus on the case description, taking into account the results of the diagnostic tests performed.
In line 45, the authors make reference to the progression of the disease over the preceding months. Could a more detailed description of the severity of symptoms be provided, with particular reference e.g. to the severity of pain and the frequency of vomiting?
Furthermore, could the CT image be described in more detail?
Finally, references should be written according to the journal's guidelines.
Thank you
Author Response
Greetings, and gratitude is extended for the time invested in providing feedback on our article.
We have given due consideration to your comments and have endeavoured to make the temporality of our patient's symptoms clearer, as well as providing a clearer explanation of the correlation between the symptoms, clinical examinations and diagnosis.
It is our sincere hope that these revisions have met your expectations.
It is our hope that our article will be published in this journal.

Round 2
Reviewer 1 Report
Comments and Suggestions for Authors
The authors have addressed all of my concerns
Comments on the Quality of English Language-